# Antimicrobial Resistance in Commensal Bacteria from Large-Scale Chicken Flocks in the Dél-Alföld Region of Hungary

**DOI:** 10.3390/vetsci12080691

**Published:** 2025-07-24

**Authors:** Ádám Kerek, Ábel Szabó, Franciska Barnácz, Bence Csirmaz, László Kovács, Ákos Jerzsele

**Affiliations:** 1Department of Pharmacology and Toxicology, University of Veterinary Medicine, István utca 2, H-1078 Budapest, Hungary; szabo.abel@student.univet.hu (Á.S.); barnacz.franciska@student.univet.hu (F.B.); csirmaz.bence@student.univet.hu (B.C.); jerzsele.akos@univet.hu (Á.J.); 2National Laboratory of Infectious Animal Diseases, Antimicrobial Resistance, Veterinary Public Health and Food Chain Safety, University of Veterinary Medicine, István utca 2, H-1078 Budapest, Hungary; kovacs.laszlo@univet.hu; 3Department of Animal Hygiene, Herd Health and Mobile Clinic, University of Veterinary Medicine, István utca 2, H-1078 Budapest, Hungary; 4Poultry-Care Kft., Lehel út 21, H-5052 Újszász, Hungary

**Keywords:** antimicrobial resistance, poultry, chickens, *Escherichia coli*, *Staphylococcus*, *Enterococcus*, MDR, Hungary, minimum inhibitory concentration

## Abstract

The rise of antimicrobial resistance (AMR) has become a critical concern for global public health. This study evaluates the antimicrobial susceptibility of commensal *Staphylococcus* spp., *Enterococcus* spp., and *Escherichia coli* strains isolated from large-scale poultry flocks in the Dél-Alföld area of Hungary. A total of 145 isolates were analyzed, using minimum inhibitory concentration (MIC) testing against 15 antimicrobial agents. Multidrug resistance was observed in 43.9% of *Staphylococcus* spp., 28.8% of *Enterococcus* spp., and 75.6% of *E. coli* isolates. Our findings highlight the importance of regional AMR surveillance, the One Health approach, and effective biosecurity and hygiene measures in intensive poultry production systems.

## 1. Introduction

Today, antimicrobial resistance (AMR) stands out as a critical concern for both human and veterinary healthcare sectors. Extensive application of antimicrobials, especially at subtherapeutic levels, has played a key role in driving the development and dissemination of resistance within poultry farming. Historically, these substances were used to enhance feed conversion efficiency; however, such applications have since been banned in the European Union and the United States [1,2]. In 2019, AMR was implicated in approximately 4.95 million deaths worldwide, with 1.27 million fatalities directly linked to infections caused by resistant bacteria, emphasizing the severity of its global burden [3]. In the United States alone, 2.8 million AMR-related infections occurred in 2019, resulting in 35,000 deaths [4]. Several countries, including Canada and members of the European Union, have implemented surveillance programs to address the growing threat of AMR [5,6].

Given the widespread dissemination of resistance, there is increasing emphasis on antibiotic-free production systems and alternative strategies such as probiotics [7], plant extracts [8,9], antimicrobial peptides [10], and novel therapeutic agents [11,12,13,14]. Subtherapeutic antibiotic use alters gut microbial composition in a compound-dependent manner [15]. Consequently, antibiotic-free poultry products are often marketed as safer alternatives for human consumption. This is supported by studies showing reduced carriage of antibiotic-resistant bacteria [16]. Research has further demonstrated that sequential exposure to antibiotics can lead to cumulative resistance, fostering the emergence of multi-drug-resistant isolates [17,18].

Antibiotics and their alternatives exert both direct and indirect effects on microbial communities, thereby influencing AMR dynamics [19]. Some antibiotics previously employed for growth promotion—such as avilamycin and bambermycin—have been associated with cross-resistance to antibiotics critical for human medicine, contributing to co-selection phenomena [20,21,22]. Therefore, how antimicrobials are used in poultry farming can significantly influence both animal welfare and public health risks, especially concerning the transmission of resistant pathogens.

Understanding resistance development requires close examination of commensal bacterial populations in poultry. *Staphylococcus* species are widespread in the environment and include both coagulase-positive and coagulase-negative isolates within poultry flocks [23]. *Staphylococcus aureus* is associated with multiple poultry diseases, including yolk sac infections (omphalitis), respiratory illness (pneumonia), and joint inflammation (arthritis) [24]. The presence of methicillin-resistant *S. aureus* (MRSA) in poultry is particularly concerning, as multiple studies have confirmed the dissemination of multidrug-resistant isolates [25].

Similarly, *Enterococcus* species—such as *Enterococcus faecalis*—are natural inhabitants of the poultry gut and may exhibit intrinsic resistance to frontline antimicrobials, including β-lactams and aminoglycosides [26]. Vancomycin-resistant *Enterococcus* (VRE) isolates raise substantial concern [27], given their potential resistance to antibiotics critical in human medicine [28,29], and the risk of transfer to the human population via the food chain [30,31].

*Escherichia coli* is a ubiquitous commensal bacterium within the gastrointestinal tract of various warm-blooded animals, including livestock species such as poultry [32], swine [33], and cattle [34]. It plays an important role in microbial ecology and is commonly used as an indicator organism in antimicrobial resistance monitoring programs across both veterinary and public health sectors [35]. Although most isolates are harmless commensals, some isolates may acquire pathogenic traits. Extraintestinal pathogenic *E. coli* (ExPEC) [36], including avian pathogenic *E. coli* (APEC), are key contributors to disease burden and economic losses in poultry farming [37,38]. Different livestock production systems, including conventional, organic, and antibiotic-free approaches, exhibit varying levels of *E. coli* resistance, with conventional systems using antibiotics often showing higher resistance rates [39,40,41]. In human medicine, infections caused by *E. coli* are often treated with cephalosporins, quinolones, aminoglycosides, and sulfonamides; however, several studies have reported multidrug resistance among commensal and APEC isolates isolated from poultry [42,43]. Commensal bacteria in poultry can act as important reservoirs and vectors of antimicrobial resistance genes, facilitating their horizontal transfer to other bacteria, including potential pathogens of both veterinary and human clinical relevance. This gene exchange can occur via mobile genetic elements such as plasmids, integrons, and transposons, increasing the risk of treatment failures and the dissemination of multidrug resistance across the food chain and the environment [44]. This underlines that antimicrobial resistance in commensal bacteria can have significant implications for animal health and welfare, as well as production efficiency and food safety, reinforcing the importance of AMR surveillance within the One Health framework.

While numerous international studies have assessed AMR in commensal poultry bacteria, data from Hungary—particularly at the regional level—remain limited [45,46]. Therefore, the aim of our study is to provide a detailed antimicrobial resistance profile of commensal bacterial isolates isolated from large-scale chicken flocks in Hungary’s Dél-Alföld region. By establishing a detailed resistance profile, we aim to present information not only about laboratory-based AMR monitoring but also its potential impacts on flock health, welfare, and production sustainability.

## 2. Materials and Methods

### 2.1. Origin of the Samples

In this study, commensal bacterial strains were isolated from fecal samples obtained from large-scale chicken farms in the Dél-Alföld region between 2022 and 2023. The isolates included *Staphylococcus* spp. strains, *Enterococcus* spp. strains, and *Escherichia coli* strains, all originating from domestic chickens (*Gallus gallus domesticus*).

Sample collection was performed on a total of three chicken farms in Dél-Alföld region, with these three farms selected through random sampling to ensure representativeness. From each farm, 15 tracheal and 15 cloacal specimens were obtained using Amies transport swabs without charcoal, featuring standard aluminum shafts (Biolab Zrt., Budapest, Hungary). Both the selection of the farms and the individual animals for sampling were carried out randomly, ensuring the generalizability of the findings. The birds included in the study were clinically healthy at the time of sampling, without any apparent signs of infection or illness.

Sample collection was carried out with the consent of the farms, in collaboration with the Department of Animal Hygiene and Herd Health, using samples taken by veterinary professionals during routine diagnostic procedures, in accordance with veterinary best practices. All farms participated anonymously. To support a One Health approach, human resistance data were also included for comparative purposes. These data were provided by the National Public Health and Pharmaceutical Center, with the necessary approvals from the Chief Medical Officer. Participating in farms voluntarily and anonymously consented to the survey, and data usage permissions were restricted to regional-level geographic information only. Although the number of animal isolates was lower compared to human isolates, this comparison was included to support the One Health perspective, highlighting potential trends across sectors. The limitations due to unequal sample sizes are acknowledged, and interpretation should be approached with caution.

Bacterial isolation was performed at the Microbiology Laboratory of the Department of Pharmacology and Toxicology, University of Veterinary Medicine Budapest. For bacterial isolation, selective and differential media were used to support the growth and preliminary identification of the target genera. *Staphylococcus* spp. isolates were cultured on CHROMagar™ Staph aureus (Chembio Fejlesztő Kft., Budapest, Hungary), *E. coli* isolates on ChromoBio^®^ Coliform agar (Biolab Zrt.) [47], and *Enterococcus* spp. isolates on m-*Enterococcus* modified agar (Merck KGaA, Darmstadt, Germany). Characteristic colony morphologies were used for preliminary selection. *Staphylococcus aureus* typically formed mauve-colored colonies on CHROMagar™ Staph aureus; *E. coli* appeared as blue-violet colonies on ChromoBio^®^ Coliform agar; and *Enterococcus* species produced red or pink colonies on m-*Enterococcus* modified agar. The species identification of the strains was determined using MALDI-TOF mass spectrometry (Flextra-LAB Kft., Budapest, Hungary) and Biotyper software version 12.0 (Bruker Daltonics GmbH, Bremen, Germany, 2024) [48]. The isolated colonies were subcultured on tryptic soy agar (Biolab Zrt.) and stored as pure cultures in Microbank™ vials (Pro-Lab Diagnostics, Richmond Hill, ON, Canada) at −80 °C until further use.

Each sample was assigned a unique identifier and accompanied by metadata including the source organ (trachea or cloaca), production type (breeding, layer, or broiler), bird age (juvenile or adult), and flock size category (5001–50,000; 50,001–100,000; >100,000).

### 2.2. Determination of Minimum Inhibitory Concentration (MIC)

Stock solutions of antimicrobial agents (Merck KGaA, Darmstadt, Germany) were prepared in line with CLSI protocols [49]. Amoxicillin and its clavulanic acid combination (2:1 ratio) were reconstituted in phosphate buffer (0.01 mol/L, pH 7.2), while imipenem was dissolved in phosphate buffer at pH 6.0 (0.1 mol/L). Ceftriaxone, doxycycline, spectinomycin, neomycin, colistin, tiamulin, tylosin, lincomycin, and vancomycin were prepared using distilled water. For the trimethoprim–sulfamethoxazole mixture (1:19 ratio), sulfamethoxazole was dissolved in heated water with a few drops of 2.5 mol/L NaOH, and trimethoprim in 0.05 mol/L HCl. Enrofloxacin was solubilized in distilled water with 1 mol/L NaOH, while florfenicol was dissolved in a 95% ethanol–distilled water mixture. The final concentration of the stock solutions was 1024 µg/mL, adjusted for the purity provided by the manufacturer.

Susceptibility testing was performed by minimum inhibitory concentration MIC determination following Clinical Laboratory Standards Institute CLSI methodology. When available, clinical breakpoints were interpreted according to CLSI guidelines [49]; otherwise, published literature breakpoints were used for *Staphylococcus* spp. in the case of imipenem [50], tylosin [51] and tiamulin [52], for *Enterococcus* spp. in the case of neomycin [53], tylosin [54] and *E. coli* in the case of neomycin [55], amoxicillin–clavulanic acid [56], spectinomycin [56], and colistin [57].

Bacterial isolates preserved in Microbank™ vials were revived by inoculation into 3 mL of cation-adjusted Mueller–Hinton broth (CAMHB) and incubated at 37 °C for 18–24 h prior to antimicrobial susceptibility testing. MIC determination was performed using 96-well microtiter plates (VWR International Ltd., Debrecen, Hungary). Each well—except those in the first column—was filled with 90 µL of CAMHB. Working antibiotic solutions (512 µg/mL) were prepared by a 1:1 dilution of the stock solutions in CAMHB, and 180 µL of this solution was added to the first column wells. Two-fold serial dilutions were performed by transferring 90 µL to the second column, mixing thoroughly, and continuing through column 10. The excess volume was discarded with the pipette tip after the final dilution, resulting in 90 µL per well.

Each row represented one bacterial isolate. Bacterial suspensions were standardized to a 0.5 McFarland turbidity using a nephelometer (CheBio Fejlesztő Kft., Budapest, Hungary) with the aid of a separate helper plate. From column 11 backward, 10 µL of each suspension was inoculated into the wells. Column 11 was designated as the growth control (containing broth and bacterial suspension), while column 12 functioned as the sterility control (containing broth only). Plates were incubated at 35 ± 2 °C for 18–24 h.

MIC values were read using a SWIN automated MIC reader (CheBio Fejlesztő Kft., Budapest, Hungary) and analyzed with the VIZION system. Reference isolates included *S. aureus* (ATCC 23235), *E. faecalis* (ATCC 29212), and *E. coli* (ATCC 25922).

### 2.3. Statistical Analysis

Statistical evaluations were performed using R software version 4.1.0. To explore resistance associations among the tested antibiotics, we employed Spearman’s rank correlation. Results were visualized in a heatmap, with varying color intensities indicating the strength of the correlations.

To assess the multivariate structure of the samples, we conducted principal component analysis (PCA), and based on the first two principal components, samples were grouped into three clusters using the K-means clustering algorithm. These clusters reflect differences in resistance profiles among the isolates.

Bacterial isolates were defined as multidrug-resistant (MDR) when resistance was observed against antimicrobial agents belonging to three or more different antibiotic classes. To predict MDR status, a decision tree model (implemented using sklearn) was constructed to identify the most influential predictors of resistance.

A weighted network graph was constructed using the Networkx package to visualize co-resistance patterns. In this graph, edges represented the number of isolates exhibiting resistance to both connected antibiotics, while node sizes reflected the total number of resistant isolates for each individual antibiotic. Edge thickness indicated the relative frequency of co-resistance between antibiotic pairs.

To model the stochastic background, a Monte Carlo simulation was performed. We generated 1000 random datasets that maintained the same number of antibiotics and isolates, but randomized resistance patterns. For each iteration, the number of MDR-classified isolates was counted and plotted as a distribution.

## 3. Results

A total of 41 *Staphylococcus* spp. isolates (Appendix A) originating from the Dél-Alföld region were analyzed for resistance against nine antibiotics based on established clinical breakpoints. The data were evaluated using multivariate statistical approaches, machine learning techniques, and advanced visualization methods. Based on MALDI-TOF analysis, these 41 *Staphylococcus* isolates included 32 *Staphylococcus aureus* ssp. *aureus*, 5 *Staphylococcus delphini*, and 4 *Staphylococcus gallinarum*. For *Enterococcus* spp. (Appendix A) isolates (*n* = 59), resistance was assessed against ten different antimicrobial agents. According to MALDI-TOF identification, the *Enterococcus* isolates consisted of 21 *Enterococcus faecium*, 26 *Enterococcus faecalis*, 6 *Enterococcus durans*, 1 *Enterococcus mundtii*, 3 *Enterococcus gallinarum*, and 2 *Enterococcus hirae*. In the case of *E. coli* (*n* = 45) isolates (Appendix A), MIC values were determined and analyzed for eleven antibiotics. All 45 isolates were identified as *E. coli*. This section will first describe the results for *Staphylococcus* spp., then *Enterococcus* spp. and finally *E. coli*. Table 1 summarizes the species-level identification and isolate counts of *Staphylococcus* spp., *Enterococcus* spp., and *E. coli* recovered from chickens. Identification was performed using MALDI-TOF MS, which revealed a predominance of *Staphylococcus aureus* ssp. *aureus* among staphylococcal isolates and *E. faecalis* and *E. faecium* among enterococci.

### 3.1. Staphylococcus spp. Isolates

A Spearman correlation matrix was constructed to examine resistance patterns. According to the resulting heatmap (Figure 1), the strongest positive correlations were observed between amoxicillin and amoxicillin–clavulanic acid (0.54) and between amoxicillin and tiamulin (0.36). No resistance was detected against vancomycin or imipenem; therefore, these agents were excluded from the correlation analysis.

PCA revealed that the isolates formed three clearly distinguishable clusters. K-means clustering confirmed that the *Staphylococcus* spp. isolates could be grouped based on distinct resistance profiles (Appendix A), suggesting differences in isolates origin or antimicrobial exposure history.

MDR isolates were defined as isolates resistant to at least three antimicrobial agents. According to the decision tree analysis, resistance to enrofloxacin and tiamulin were the primary predictors of MDR status.

A weighted network graph was generated to visualize co-resistance between antimicrobials (Appendix A). Node size represented the number of resistant isolates per compound, while edge thickness indicated the frequency of dual resistance between pairs of antimicrobials. The most densely connected pairs were potentiated sulfonamide–enrofloxacin, potentiated sulfonamide–tiamulin, and enrofloxacin–tiamulin, suggesting potential cross-resistance mechanisms.

Monte Carlo simulation with 1000 iterations was used to model stochastic resistance patterns, maintaining the sample size (*n* = 41). On average, 31.8 MDR isolates were expected per simulation (Appendix A), serving as a baseline to evaluate whether the observed MDR frequency exceeded what would be expected by random chance.

Based on MIC values, a frequency table was compiled (Table 2), and the proportions of resistant and susceptible isolates were visualized according to clinical breakpoints (Figure 2). The highest level of resistance (97.6%) was observed for potentiated sulfonamides. While amoxicillin resistance was detected in 61% of isolates, the reduced resistance rate of 39% for amoxicillin–clavulanic acid suggests beta-lactamase production in a subset of isolates. Notably, resistance to enrofloxacin was observed in 73.2% of isolates, raising significant concern. In contrast, all isolates remained susceptible to imipenem and vancomycin, which are reserved for critical human medical use.

### 3.2. Enterococcus spp. Isolates

The results for *Enterococcus* spp. were similar to those for *Staphylococcus* spp. Correlation analysis revealed the strongest positive association between amoxicillin and amoxicillin–clavulanic acid resistance (0.81), while vancomycin also showed a weaker yet statistically significant correlation with these beta-lactam antibiotics (Figure 3). Among the remaining agents, only weak or negligible correlations were observed.

PCA resulted in three well-separated clusters (Appendix A), each reflecting distinct resistance profiles. One cluster predominantly contained MDR isolates, while the other two were characterized by more limited resistance patterns. These differences may reflect underlying biological variation or differences in antimicrobial exposure history.

A decision tree model was constructed to predict MDR status. Resistance to tylosin, florfenicol, and doxycycline emerged as the most important predictors. The model’s logic suggests that MDR status can be reliably inferred from resistance to a small number of key antimicrobial agents.

The network of co-resistance relationships among resistant isolates was visualized using a weighted graph (Appendix A). The largest node was associated with florfenicol, which also exhibited the strongest connections—particularly with tylosin and neomycin—suggesting possible co-selection or cross-resistance mechanisms.

Monte Carlo simulations predicted that, under random distribution of resistance, an average of 50.4 MDR isolates would be expected per simulation (Appendix A). However, only 18 MDR isolates were observed in the actual dataset, a significantly lower number. This finding indicates that the emergence of MDR is likely not random, but influenced by selective pressure, controlled antimicrobial use, or other regulatory factors.

Based on the MIC values, we compiled a frequency table (Table 3) and determined the proportion of resistant and susceptible isolates for each antimicrobial agent according to clinical breakpoints (Figure 4). The highest resistance rate was observed for florfenicol (54.2%). All isolates remained susceptible to imipenem, which is reserved for critical human medical use. In contrast, vancomycin resistance was identified in 11.9% of isolates, raising concern due to the clinical importance of this agent. A majority of isolates (83.1%) retained susceptibility to amoxicillin.

### 3.3. Escherichia coli Isolates

The results for *E. coli* did not differ wildly from those for *Staphylococcus* spp. or *Enterococcus* spp. Spearman’s correlation analysis revealed the strongest positive associations between enrofloxacin and amoxicillin resistance (0.55), and between spectinomycin and amoxicillin–clavulanic acid (0.50) (Figure 5). In contrast, florfenicol showed weak or even negative correlations with several other antimicrobials.

PCA followed by K-means clustering, effectively separated the isolates into three distinct clusters (Appendix A). One cluster predominantly contained isolates with high MDR profiles, while the other two included less resistant isolates. These findings suggest the existence of biologically and epidemiologically distinct subpopulations within the *E. coli* isolates.

The decision tree model identified doxycycline, colistin, and enrofloxacin resistance as the most important predictors of MDR status. This suggests that resistance to these agents is strongly associated with broader multidrug resistance patterns.

A weighted co-resistance network graph (Appendix A) was used to illustrate the frequency of simultaneous resistance occurrences. The strongest connections were observed among florfenicol, neomycin, and enrofloxacin, indicating potential co-selection and common resistance mechanisms.

Monte Carlo simulations estimated that 42.5 MDR isolates would be expected under a random distribution model. The actual observed number—34 MDR isolates—was significantly lower (Appendix A), suggesting that MDR development in these isolates is likely influenced by structured selective pressures rather than random chance.

MIC data were used to create a frequency distribution (Table 4), and susceptibility profiles were determined for each antimicrobial based on established clinical breakpoints (Figure 6). Florfenicol exhibited the highest resistance rate at 82.2%. Although 44.4% of isolates were resistant to amoxicillin, resistance to the amoxicillin–clavulanic acid combination was notably lower at 6.7%, suggesting that many isolates may produce beta-lactamase enzymes.

### 3.4. Multidrug-Resistant (MDR), Extensively Drug-Resistant (XDR), and Pan-Drug-Resistant (PDR) Isolates

Significant differences were observed in the antimicrobial resistance (AMR) profiles of different bacterial species isolated from chickens in the Dél-Alföld region. Among *Staphylococcus* spp. isolates (*n* = 41), 18 (43.9%) were classified as MDR, and in addition, three XDR and five PDR isolates were identified—representing the most extreme resistance patterns among the examined species. In contrast, among *Enterococcus* spp. isolates (*n* = 59), 17 (28.8%) were MDR, only one isolate was XDR, and no PDR isolates were detected. For *E. coli* isolates (*n* = 45), 75.6% (34 isolates) were MDR, but none exhibited XDR or PDR phenotypes (Figure 7). These findings indicate that while *Staphylococcus* spp. isolates exhibited a lower overall MDR prevalence, they harbored the most critical resistance forms. Conversely, *E. coli* isolates primarily displayed broad, though not absolute, resistance.

### 3.5. Comparison with Regional Human Resistance Data

The resistance levels of *Staphylococcus* spp. isolates from chickens exceeded those observed in human clinical isolates for most tested antimicrobials (Figure 8). The most striking differences were found in the rates of resistance to potentiated sulfonamides (97.6% vs. 0.6%) and fluoroquinolones (73.2% vs. 9.5%). Resistance to macrolides was comparable between the animal and human isolates (approximately 31%). No vancomycin-resistant isolates were detected in either group.

In the case of *Enterococcus* spp. isolates (Figure 9), resistance patterns differed markedly between animal and human sources. Aminopenicillin resistance in poultry isolates was 16.9%, while it was undetectable in human *E. faecalis* isolates but extremely high (97.6%) in *E. faecium*. Aminoglycoside resistance levels were comparable in poultry-derived *Enterococcus* spp. (32.2%) and human *E. faecalis* (29.0%) but higher in human *E. faecium* (56.2%). Vancomycin resistance in poultry isolates was 11.9%, lower than the rates seen in human isolates (*E. faecalis* 20.0%, *E. faecium* 32.0%).

For *E. coli* isolates (Figure 10), resistance was generally higher in chicken-derived isolates compared to those from human sources. Aminoglycoside resistance was particularly elevated in poultry isolates (57.8% vs. 7.7%), as were resistance rates for fluoroquinolones (44.4% vs. 17.8%) and potentiated sulfonamides (55.6% vs. 22.0%). Resistance to aminopenicillins was nearly identical between the two sources. Notably, resistance to amoxicillin–clavulanic acid was lower in poultry-derived isolates (6.7%) than in human isolates (17.0%).

## 4. Discussion

The analysis of *Staphylococcus* spp. isolates derived from chickens in Hungary’s Dél-Alföld region revealed notable multidimensional differences in antimicrobial resistance. In total, 41 *Staphylococcus* spp. isolates were examined. Strong positive correlation between amoxicillin and amoxicillin–clavulanic acid reflected their similar mechanisms of action and suggest [58] the presence of β-lactamase producers, consistent with findings in larger Hungarian poultry studies (*n* = 227 isolates with 58–61% resistance rates). Additionally, the correlation between enrofloxacin and tiamulin suggested possible co-exposure or sequential use, a pattern previously reported in turkey flocks. Cluster analysis identified three distinct resistance profiles, likely reflecting varying antibiotic usage across farms [59,60,61]. These findings align with the decision tree model results, where enrofloxacin, tiamulin, and potentiated sulfonamides emerged as key predictors of MDR status. Beyond these laboratory-based findings, understanding resistance profiles in commensal bacteria is essential because these bacteria can serve as reservoirs of resistance genes, potentially impacting animal health, welfare, and overall production performance.

The weighted network graph revealed complex cross-resistance mechanisms, such as potentiated sulfonamide–tiamulin and enrofloxacin–tiamulin, also observed in larger poultry datasets [46]. Monte Carlo simulation indicated that an average of 31.8 MDR isolates would be expected under random resistance patterns for the given sample size. This serves as a benchmark for interpreting the observed MDR prevalence, suggesting that the actual occurrence likely exceeds what would be expected by chance. Overall, the results underscore the importance of targeted surveillance of antibiotic resistance in *Staphylococcus* spp. isolates from the Dél-Alföld region to prevent the emergence and dissemination of MDR isolates.

Amoxicillin resistance was observed in 61% of isolates, closely aligning with the 51.2% resistance to penicillins reported by Kim et al. [62] and closely matching the 58.6% resistance reported in Hungarian poultry by Szabó et al. [58]. In contrast, enrofloxacin resistance was relatively higher (73.2%) than the 55.1% reported in the same context [58], considerably higher than the 33.9% ciprofloxacin resistance reported by Kim et al. [62]. It likely reflects intensive and prolonged fluoroquinolone use in broiler production, leading to stepwise selection and accumulation of target-site mutations in *gyrA/parC* [63].

The lower observed resistance (39.0%) to amoxicillin–clavulanic acid, compared to amoxicillin alone, is consistent with the presence of β-lactamase–producing, but clavulanate-inhibited, strains, as also reported by Szabó et al. [58]. Although amoxicillin–clavulanic acid is not approved for use in poultry due to the lack of maximum residue limits (MRL). Comparative literature on ceftriaxone and imipenem resistance in poultry-associated *Staphylococcus* spp. is scarce; however, all isolates in this study were susceptible to imipenem—a drug reserved for human medicine—highlighting its critical importance.

For doxycycline, our 51.2% resistance rate is in line with the 74.4% from Szabó et al. and higher than the 38.8% tetracycline resistance noted by Kim et al. [58,62], reflecting widespread use of tetracyclines in poultry. Our results are consistent with Miranda et al., who reported a similar resistance level of 58.4% [64]. All isolates were susceptible to vancomycin in the present study. However, Mkize et al. detected vancomycin resistance in 14% of *Staphylococcus* spp. strains isolated from fecal samples and in 61.9% of those from slaughterhouse samples [65].

The *Enterococcus* spp. isolates (*n* = 59) from the Dél-Alföld region exhibited a notable MDR prevalence of 30%, with particularly high resistance observed against florfenicol, tylosin, and neomycin. Both decision tree and network analyses confirmed that these antibiotics were key determinants in the development of MDR status. According to the Monte Carlo simulations, the observed MDR rate was significantly lower than expected by chance, indicating a controlled selective pressure or heterogeneous antibiotic exposure across isolates. These findings underscore the importance of continuous monitoring, particularly for veterinary antibiotics such as florfenicol and neomycin, which are widely used and whose resistance may pose future therapeutic challenges. Such challenges are not only relevant to the treatment of clinical infections but may also compromise animal welfare and productivity in poultry farming systems.

Resistance to amoxicillin was detected in 16.9% of isolates. This rate is notably lower than the 43–63% range reported by Bekele et al. in conventional broiler farms [66], but aligns with the much lower 6.25% penicillin resistance observed in free-range systems by Lemsaddek et al. [67]. This difference likely arises from distinct antibiotic usage patterns and farming systems: intensive indoor poultry operations tend to employ antimicrobials more routinely and at higher rates compared to free-range or smaller-scale farms [68,69].

Resistance to amoxicillin–clavulanic acid was 11.9%, falling within the broad range of prior findings: from 1.47% in livestock by Pesavento et al. [70] to 73.3% in poultry by Ayeni et al. [71]. These discrepancies may stem from regional differences in antimicrobial stewardship and the prevalence of β-lactamase-producing strains.

Neomycin resistance was observed in 32.2%, compared to 83% reported by Lanza et al. [72], suggesting differing neomycin usage patterns across regions and flocks. Doxycycline resistance was 18.6% in our data, which was lower than Bekele et al.’s 26–41% [66] and Noh et al.’s 58.2% [73] but similar to Schwaiger et al.’s 39.7% in organic broilers [17], reinforcing that production system (organic vs. conventional) significantly impacts resistance rates, likely due to lower antibiotic pressure in extensive systems.

Strikingly, florfenicol resistance reached 54.2%, whereas Schwaiger et al. reported no resistance in either setting [17], and Karunarathna et al. found only 0.4% [74]. Kim et al. reported a 14.3–18.7% range [75]. This suggests an elevated florfenicol use or co-selection pressures (e.g., plasmid-mediated multi-resistance) within the region studied.

Tylosin resistance (35.6%) was moderate compared to Kim et al.’s 53–63.6% [75]. Tylosin, commonly used in poultry respiratory infections, may be less used in Hungarian flocks, explaining lower prevalence.

Enrofloxacin resistance (13.6%) is significantly below the 83.3% reported by Oliveira et al. [76], yet closer to Karunarathna et al.’s 5.1% [74]. Large differences in enrofloxacin exposure—intensive use in some countries versus restricted or prudent use policies in others—likely account for variation.

Potentiated sulfonamide resistance (15.3%) is lower than Makarov et al.’s 32.9–36.6% [77] but higher than Karunarathna et al.’s 7.4% [74], further highlighting geographic and management differences in antibiotic application.

Vancomycin resistance (11.9%) is concerning. Previous studies in poultry detected none by Schwaiger [17] and Semedo-Lemsaddek [67] or very low levels (1.9–4.4%) by Karunarathna [74], Pesavento [70], and Roy [78]. In contrast, Bekele et al. found 15–66% [66], and Ayeni et al. 65% in conventional farms [71]. This elevated rate may reflect historic use of avoparcin (a vancomycin analog) in the region, which is well documented to select for VRE in poultry [79]. Additionally, our results may include intrinsic resistance of species like *E. gallinarum* which carry the *vanC* gene [80], further contributing to the observed prevalence.

The *E. coli* isolates (*n* = 45) from the Dél-Alföld region exhibited particularly high levels of antimicrobial resistance. The MDR prevalence exceeding 75% indicates a heavily burdened bacterial population, especially with resistance to florfenicol, doxycycline, and neomycin. Results from decision tree and clustering analyses showed that MDR isolates could be distinctly separated based on their resistance profiles, highlighting the potential utility of these models in rapid diagnostic support. Monte Carlo simulation confirmed that the observed MDR presence cannot be explained by random distribution alone, suggesting that intensive antibiotic usage or horizontal gene transfer is likely contributing to the resistance patterns. These findings underscore the need for stricter monitoring and targeted application of antibiotics in poultry production systems.

Resistance to amoxicillin was 44.4%. This aligns with Hassan et al.’s 32% resistance [81] to ampicillin in poultry-associated *E. coli*. However, Kaushik et al. reported much higher levels, 89.4% penicillin resistance [82], reflecting variability in antibiotic use across geographical regions [83]. Such disparities likely result from differences in farm-level antibiotic administration practices; intensive indoor settings often employ higher frequencies of aminopenicillins compared to more restricted usage in free-range or organic systems [83]. This hypothesis aligns with meta-analytical findings showing increased resistance prevalence in intensively reared poultry.

Resistance to amoxicillin–clavulanic acid was much lower (6.7%), supporting the conclusion that a significant share of resistant isolates carry β-lactamase enzymes that clavulanic acid can inhibit. In contrast, Majewski et al. reported 84.6% resistance [84], likely reflecting unregulated β-lactamase prevalence in other settings.

Ceftriaxone resistance was 22.2%, similar to Kaushik et al.’s 28.2% [82], but Mandal et al. reported 78.1% cefotaxime resistance [85]. These discrepancies may stem from regional variations in third-generation cephalosporin usage and surveillance protocols [86].

Imipenem resistance was 11.1%, concerning for a reserve antibiotic in human medicine. Imipenem is strictly controlled in veterinary use across most regions, so any detected resistance likely arises from co-selection pressures or horizontal gene transfer rather than direct exposure. By contrast, Shaib et al. observed no resistance [86], while Moffo et al. reported 20% [87], once again underscoring variations in resistance emergence linked to differing antibiotic stewardship levels [88].

Neomycin resistance was 56.8%, compared to Majewski’s 84.6% [84]. Similar deviations are seen with gentamicin and spectinomycin, highlighting how farm-specific aminoglycoside usage and sampling methodologies influence outcomes [88].

Doxycycline resistance (53.3%) falls between Hassan et al.’s 44% [81] and Mandal et al.’s 78.1% [85]. These differences reflect diverse veterinary tetracycline use practices across countries and production systems.

Florfenicol resistance was notably high at 82.2%, while other studies reported it to be as low as 0.9% [89]. High usage of florfenicol in certain poultry sectors may select strongly for resistant *E. coli* variants, which would explain such elevated levels.

Enrofloxacin resistance was 44.4%, exceeding Hassan et al. [81] and Majewski et al. (32–34.6%) [84], reflecting continued reliance on quinolones in intensive poultry farming. Much et al. reported [89] similar resistance in conventional systems [88].

Colistin resistance was 24.4%, higher than Kempf et al.’s 0.6% [90]. This may point to the spread of mobile *mcr* genes, possibly linked to environmental contamination from other livestock systems.

Resistance to potentiated sulfonamides stood at 55.6%, consistent with Majewski’s findings [84]. Lower values reported elsewhere suggest variability in trimethoprim–sulfonamide use and resistance gene prevalence.

Our findings support the importance of routine, large-scale antimicrobial susceptibility testing with regional stratification. Such efforts provide crucial local data to inform targeted interventions. Most of the antibiotics primarily used in veterinary practice retained a high degree of effectiveness; however, the growing resistance to antimicrobials critical to public health is concerning. The recent upward trend in resistance rates sends a clear signal that strengthening a multidisciplinary approach is essential, particularly through enhanced collaboration between veterinary and public health professionals. The implementation of the One Health concept is not merely a theoretical framework but a pressing need requiring concrete action—including the development of preventive strategies, the expansion of educational programs, and intensified cross-sector communication. Only through this integrated approach can we effectively mitigate the spread of resistance and ensure long-term protection of human health. Moreover, these measures directly support animal health and welfare, while also safeguarding the sustainability of intensive poultry production.

It is important to emphasize that, despite the smaller number of animal isolates, their inclusion in cross-sectoral comparisons offers valuable preliminary insights for antimicrobial resistance monitoring. These comparisons are crucial for implementing One Health strategies, especially in regions where such integrative data are scarce. Nonetheless, the imbalance in sample sizes between sectors is a limitation, and results should be interpreted accordingly.

One notable limitation of this study is the lack of targeted genetic analyses, such as whole-genome sequencing (WGS) or targeted next-generation sequencing (NGS), which could have provided deeper insights into the specific resistance genes, their potential for horizontal transfer, and the clonal relationships among the isolates. While the phenotypic antimicrobial susceptibility testing employed here offers practical insights for veterinary decision-making and One Health surveillance, it does not allow confirmation of the genetic mechanisms underlying the observed resistance profiles. The absence of genotypic confirmation limits our ability to distinguish between intrinsic and acquired resistance and prevents detection of co-resistance or mobile genetic elements (e.g., plasmids or transposons). Moreover, without genotypic data, it is not possible to infer whether identical resistance patterns stem from clonal spread or independent selection events. Future studies incorporating genomic analyses are therefore warranted to complement our findings, provide molecular-level validation, and enhance the epidemiological interpretation of resistance trends. Such integration would also allow for better risk assessment regarding zoonotic transmission of resistance determinants.

These limitations should be carefully considered when interpreting the study results. While the phenotypic data presented offer valuable insights into resistance burden and patterns in poultry-associated commensals, they cannot provide full resolution of the underlying resistance mechanisms or traceability of resistance gene flow. Thus, the conclusions drawn from this study are most applicable at the phenotypic and surveillance level and should be complemented with molecular studies in future research frameworks.

In the medium to long term, reducing antimicrobial resistance in poultry will require strengthened biosecurity, hygiene interventions, and optimized animal rearing practice strategies that can lower disease incidence and antimicrobial use, thereby reducing selection pressure.

Complementing these practical efforts, it is equally important to develop a deeper understanding of resistance mechanisms through expanded genomic surveillance. Despite the lack of targeted genetic analyses in this study, our results provide one of the most comprehensive phenotypic resistance profiles of poultry-associated commensal bacteria in the region. This baseline offers valuable input for future molecular studies and supports the design of integrated One Health strategies aimed at curbing antimicrobial resistance.

## 5. Conclusions

This study provides detailed insights into the antimicrobial resistance profiles of commensal *Staphylococcus* spp., *Enterococcus* spp., and *E. coli* strains isolated from industrial-scale chicken flocks in the Dél-Alföld region of Hungary. The findings reveal substantial levels of multidrug resistance, especially among *E. coli* isolates, and the presence of critically resistant isolates in *Staphylococcus* spp. populations. Resistance patterns often exceeded those observed in human clinical isolates, particularly for fluoroquinolones and potentiated sulfonamides, suggesting a heightened selective pressure in the veterinary sector. These results emphasize the urgent need for region-specific AMR surveillance, prudent antimicrobial use in poultry production, and the implementation of rigorous animal hygiene and biosecurity practices. A robust interdisciplinary approach under the One Health paradigm is essential to curb the spread of resistance and safeguard both animal and public health.

If current antimicrobial usage practices remain unchanged, there is a serious risk that commensal bacterial populations will continue to evolve into reservoirs of clinically significant resistance genes. This may lead to the spillover of multidrug-resistant pathogens to humans through the food chain or the environment, as demonstrated by studies finding resistant *E. coli* and *Salmonella* on retail poultry meat and environmental samples from farms [91,92,93,94], threatening the efficacy of last-resort antibiotics and increasing the incidence of untreatable infections. By identifying these risks early, this study aims to support targeted interventions to prevent further escalation. The primary objective is to reduce selection pressure in food-producing animals, limit cross-species transmission of resistance genes, and preserve the long-term effectiveness of essential antimicrobials.

## Figures and Tables

**Figure 1 vetsci-12-00691-f001:**
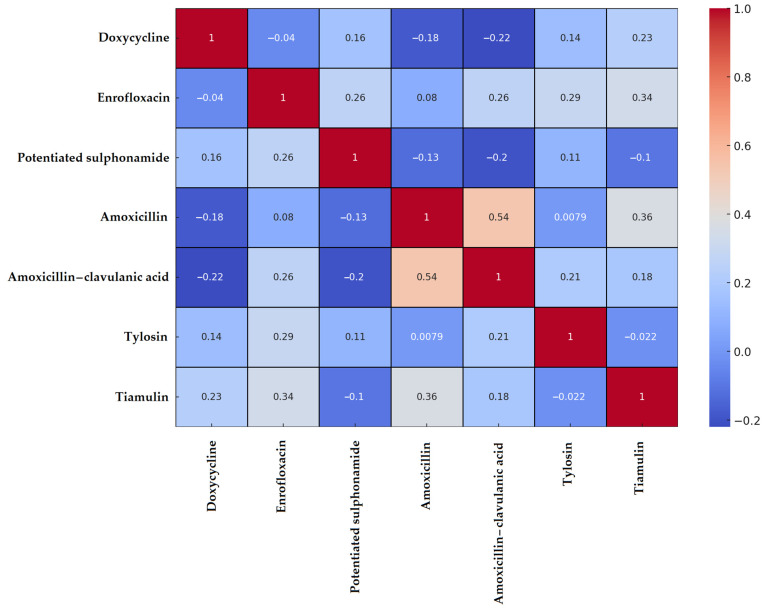
Spearman correlation heatmap of *Staphylococcus* spp. isolates (*n* = 41) isolated from chickens in the Dél-Alföld region. A positive correlation indicates a higher probability of co-occurrence of resistance to both antimicrobial agents. A correlation value close to zero suggests that resistance to the two agents develops independently. A negative correlation implies that resistance to one agent may be associated with susceptibility to the other, indicating an inverse relationship.

**Figure 2 vetsci-12-00691-f002:**
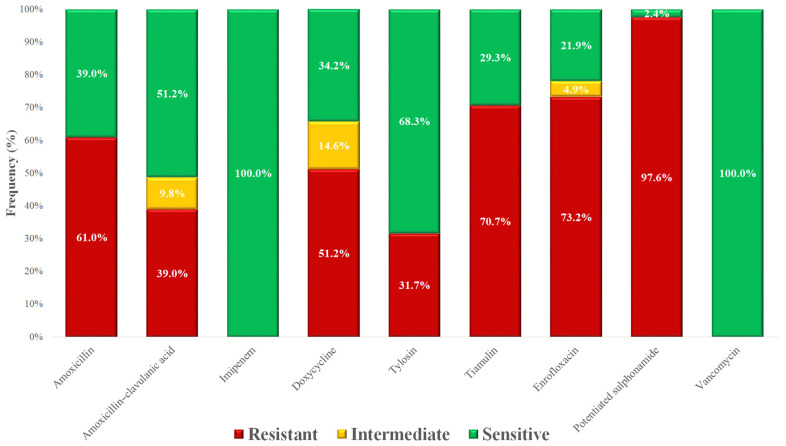
Antimicrobial resistance profile of *Staphylococcus* spp. isolates (*n* = 41) isolated from poultry in the Dél-Alföld region.

**Figure 3 vetsci-12-00691-f003:**
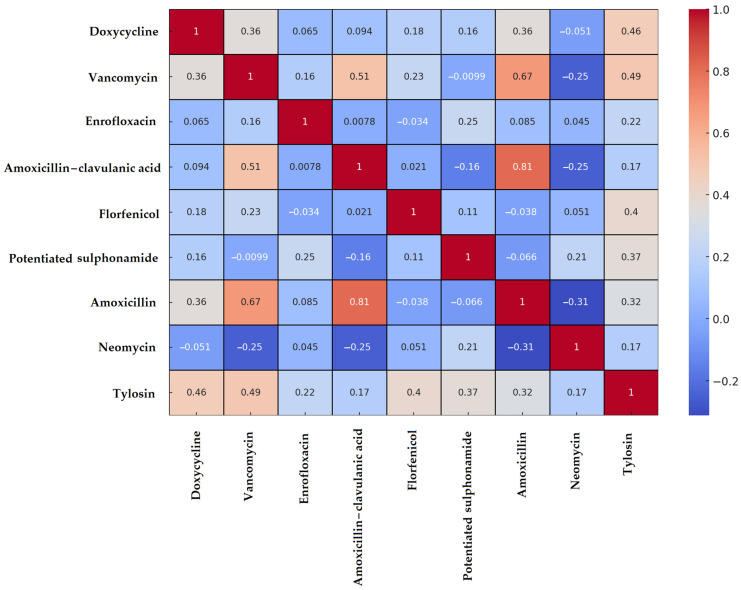
Spearman correlation heatmap of *Enterococcus* spp. isolates (*n* = 59) isolated from chickens in the Dél-Alföld region. Positive correlation values indicate co-occurrence of resistance to both agents, values near zero suggest independence, and negative correlations imply inverse associations.

**Figure 4 vetsci-12-00691-f004:**
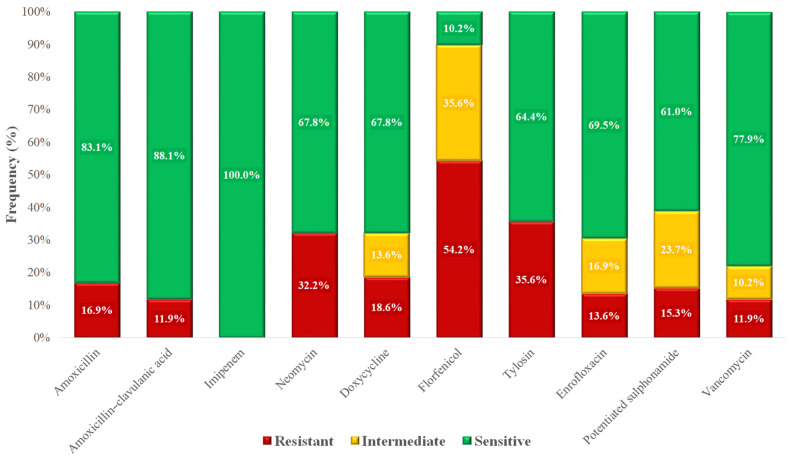
Antimicrobial resistance profile of *Enterococcus* spp. isolates (*n* = 59) isolated from chickens in the Dél-Alföld region.

**Figure 5 vetsci-12-00691-f005:**
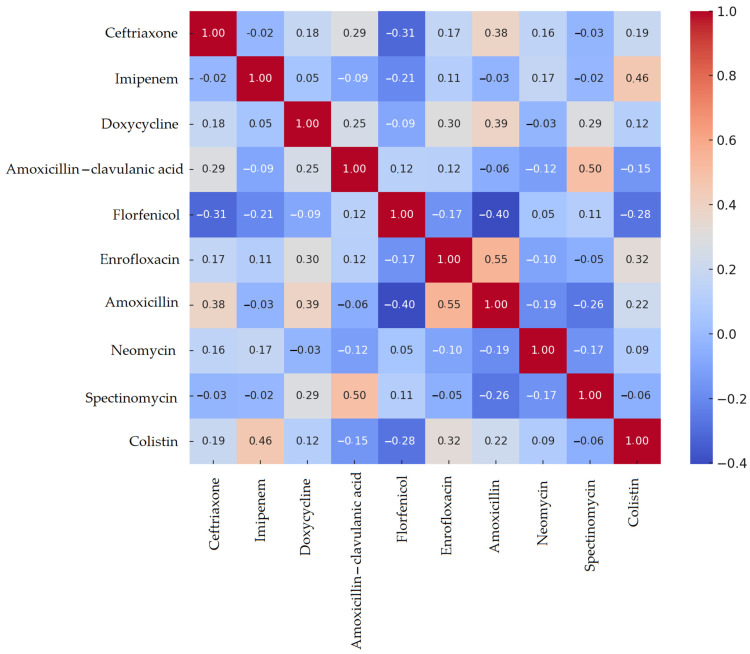
Spearman correlation heatmap of *Escherichia coli* isolates (*n* = 45) isolated from chickens in the Dél-Alföld region. Positive correlations indicate co-resistance, values near zero indicate independence, and negative values suggest inverse resistance patterns.

**Figure 6 vetsci-12-00691-f006:**
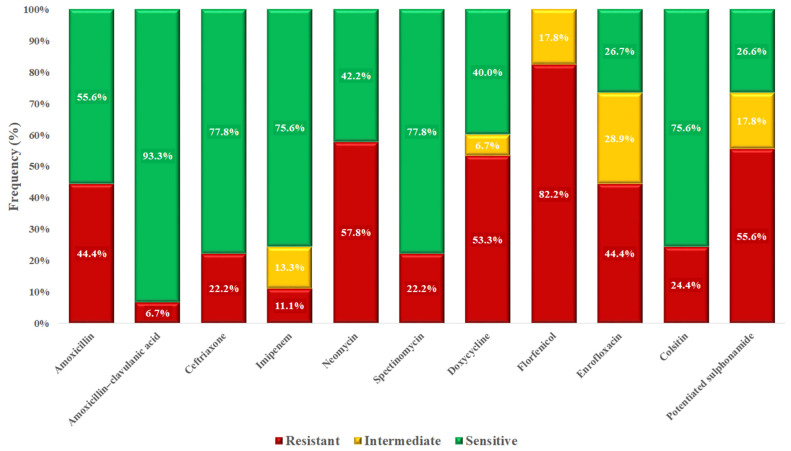
Antimicrobial resistance profile of *Escherichia coli* isolates (*n* = 45) isolated from chickens in the Dél-Alföld region.

**Figure 7 vetsci-12-00691-f007:**
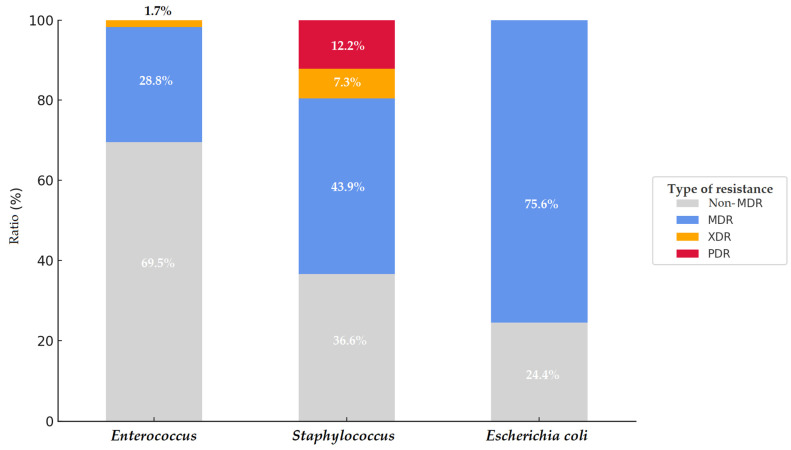
Comparative distribution of multidrug-resistant (MDR), extensively drug-resistant (XDR), and pandrug-resistant (PDR) isolates among isolates from the Dél-Alföld region.

**Figure 8 vetsci-12-00691-f008:**
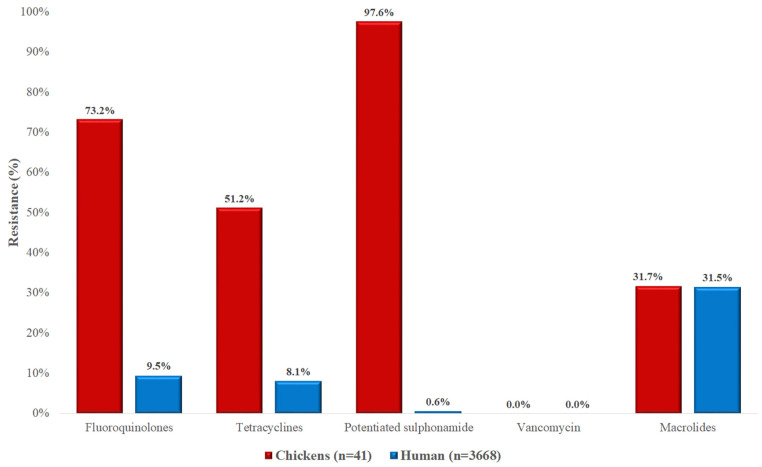
Comparison of antibiotic resistance between *Staphylococcus* spp. isolates from chickens (*n* = 41) and human sources (*n* = 3668) across different antimicrobial classes. Notably higher resistance rates were observed in poultry-derived isolates against potentiated sulfonamides, fluoroquinolones, and doxycycline.

**Figure 9 vetsci-12-00691-f009:**
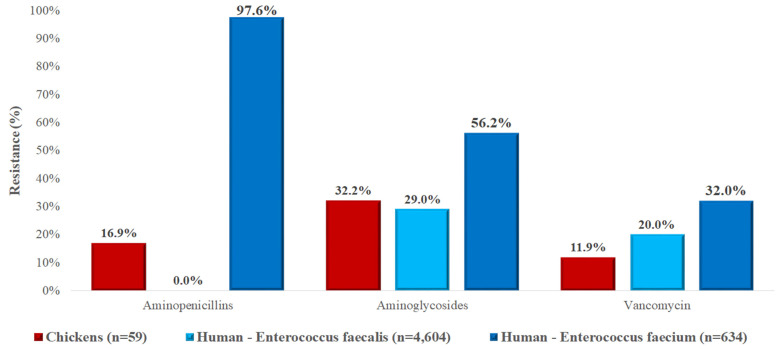
Comparison of antibiotic resistance in *Enterococcus* spp. isolates from chickens (*n* = 59) and human sources, including *Enterococcus faecalis* (*n* = 4604) and *Enterococcus faecium* (*n* = 634). Notably, resistance to aminopenicillins was extremely high in *E. faecium*, while poultry-derived isolates showed moderate resistance across all tested antimicrobial classes.

**Figure 10 vetsci-12-00691-f010:**
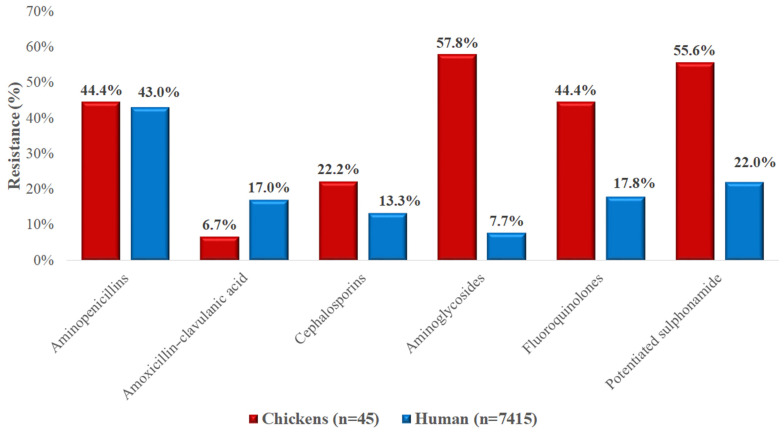
Comparison of antibiotic resistance in *Escherichia coli* isolates from chickens (*n* = 45) and human sources (*n* = 7415) across various antimicrobial classes. Poultry-derived isolates exhibited higher resistance in most categories, especially to aminoglycosides, fluoroquinolones, and potentiated sulfonamides.

**Table 1 vetsci-12-00691-t001:** Species-level identification and isolate counts of *Staphylococcus* spp., *Enterococcus* spp., and *Escherichia coli* from chickens in the Dél-Alföld region of Hungary.

Species	Number of Isolates
*Staphylococcus* spp. (*n* = 41)	
*Staphylococcus aureus* ssp. *aureus*	32
*Staphylococcus delphini*	5
*Staphylococcus gallinarum*	4
*Enterococcus* spp. (*n* = 59)	
*Enterococcus faecalis*	26
*Enterococcus faecium*	21
*Enterococcus durans*	6
*Enterococcus gallinarum*	3
*Enterococcus hirae*	2
*Enterococcus mundtii*	1
*Escherichia coli* (*n* = 45)	

**Table 2 vetsci-12-00691-t002:** Frequency table of minimum inhibitory concentration (MIC) values for antimicrobial agents with established clinical breakpoints in *Staphylococcus* spp. isolates (*n* = 41) from chickens in the Dél-Alföld region. The upper row for each compound displays the number of isolates, while the lower row shows the corresponding percentage. Vertical red lines indicate the clinical breakpoint, and green lines represent the epidemiological cutoff value (ECOFF).

Antibiotics	^1^ Breakpoint	0.001	0.002	0.004	0.008	0.016	0.031	0.063	0.125	0.25	0.5	1	2	4	8	16	32	64	128	256	512	1024	MIC_50_	MIC_90_	^2^ ECOFF
	µg/mL																								
Amoxicillin	0.5							1	4	11	8	11	2	1	1	1	0	0	0	0	1		0.5	2	0.5
								2.4%	9.8%	26.8%	19.5%	26.8%	4.9%	2.4%	2.4%	2.4%	0.0%	0.0%	0.0%	0.0%	2.4%				
^3^ Amoxicillin–clavulanic acid	1						2	1	5	13	4	13	0	3									0.25	1	0.5
							4.9%	2.4%	12.2%	31.7%	9.8%	31.7%	31.7%	7.3%											
Doxycycline	0.5	2	1	2	0	5	0	2	2	6	5	0	1	4	1	2	0	3	5				0.5	128	0.5
		4.9%	2.4%	4.9%	0.0%	12.2%	0.0%	4.9%	4.9%	14.6%	12.2%	0.0%	2.4%	9.8%	2.4%	4.9%	0.0%	7.3%	12.2%						
Enrofloxacin	4						1	2	3	2	1	2	2	9	2	9	2	5	1	1	1		8	64	0.5
							2.4%	4.9%	7.3%	4.9%	2.4%	4.9%	4.9%	22.0%	4.9%	22.0%	4.9%	12.2%	2.4%	2.4%	2.4%				
Imipenem	8							5	7	9	12	7	1										0.25	1	0.125
								12.2%	17.1%	22.0%	29.3%	17.1%	2.4%												
^4^ Potentiated sulphonamide	4												1	0	0	8	6	10	0	2	3	11	64	1024	0.25
													2.4%	0.0%	0.0%	19.5%	14.6%	24.4%	0.0%	4.9%	7.3%	26.8%			
Tylosin	64			1	0	0	1	5	0	2	5	4	1	3	5	0	1	0	1	0	0	12	4	1024	2
				2.4%	0.0%	0.0%	2.4%	12.2%	0.0%	4.9%	12.2%	9.8%	2.4%	7.3%	12.2%	0.0%	2.4%	0.0%	2.4%	0.0%	0.0%	29.3%			
Tiamulin	4								2	3	0	3	4	6	2	4	1	15	0	0	0	1	16	64	2
									4.9%	7.3%	0.0%	7.3%	9.8%	14.6%	4.9%	9.8%	2.4%	36.6%	0.0%	0.0%	0.0%	2.4%			
Vancomycin	32								1	17	10	8	5										0.5	2	2
									2.4%	41.5%	24.4%	19.5%	12.2%												

^1^ Clinical Laboratory Standard Institute (CLSI); ^2^ epidemiological cut-off value (EUCAST); ^3^ 2:1 ratio; ^4^ trimetoprim–sulphametoxazole in ratio 1:19.

**Table 3 vetsci-12-00691-t003:** Frequency table of minimum inhibitory concentration (MIC) values for antimicrobial agents with established clinical breakpoints in *Enterococcus* spp. isolates (*n* = 59) from chickens in the Dél-Alföld region. The upper row for each compound displays the number of isolates, while the lower row shows the corresponding percentage. Vertical red lines indicate the clinical breakpoint, and green lines represent the epidemiological cutoff value (ECOFF).

Antibiotics	^1^ Breakpoint	0.001	0.002	0.004	0.008	0.016	0.031	0.063	0.125	0.25	0.5	1	2	4	8	16	32	64	128	256	512	1024	MIC_50_	MIC_90_	^2^ ECOFF
	µg/mL																								
Amoxicillin	16									1	13	16	17	2	0	1	3	0	2	1	1	2	1	128	-
										1.7%	22.0%	27.1%	28.8%	3.4%	0.0%	1.7%	5.1%	0.0%	3.4%	1.7%	1.7%	3.4%			
^3^ Amoxicillin–clavulanic acid	16								1	6	20	19	4	2	0	2	1	0	4				1	16	-
									1.7%	10.2%	33.9%	32.2%	6.8%	3.4%	0.0%	3.4%	1.7%	0.0%	6.8%						
Doxycycline	16			3	1	3	0	0	0	3	0	3	6	21	8	8	1	1	1				4	16	1
				5.1%	1.7%	5.1%	0.0%	0.0%	0.0%	5.1%	0.0%	5.1%	10.2%	35.6%	13.6%	13.6%	1.7%	1.7%	1.7%						
Enrofloxacin	4							3	0	9	29	8	2	4	0	1	0	0	2	0	1		0.5	4	-
								5.1%	0.0%	15.3%	49.2%	13.6%	3.4%	6.8%	0.0%	1.7%	0.0%	0.0%	3.4%	0.0%	1.7%				
Florfenicol	8												6	21	24	5	2	1					8	16	8
													10.2%	35.6%	40.7%	8.5%	3.4%	1.7%							
Imipenem	16					1	1	0	1	0	3	14	28	9	2								2	4	4
						1.7%	1.7%	0.0%	1.7%	0.0%	5.1%	23.7%	47.5%	15.3%	3.4%										
Neomycin	1024										2	2	0	2	1	1	3	0	6	11	12	19	512	1024	256
											3.4%	3.4%	0.0%	3.4%	1.7%	1.7%	5.1%	0.0%	10.2%	18.6%	20.3%	32.2%			
^4^ Potentiated sulphonamide	64										1	6	6	8	15	8	6	3	1	1	0	4	8	128	-
											1.7%	10.2%	10.2%	13.6%	25.4%	13.6%	10.2%	5.1%	1.7%	1.7%	0.0%	6.8%			
Tylosin	8										5	15	12	6	0	0	4	0	0	0	4	13	2	1024	-
											8.5%	25.4%	20.3%	10.2%	0.0%	0.0%	6.8%	0.0%	0.0%	0.0%	6.8%	22.0%			
Vancomycin	32								1	1	8	12	16	8	4	2	0	0	0	0	0	7	2	1024	4
									1.7%	1.7%	13.6%	20.3%	27.1%	13.6%	6.8%	3.4%	0.0%	0.0%	0.0%	0.0%	0.0%	11.9%			

^1^ Clinical Laboratory Standard Institute (CLSI); ^2^ epidemiological cut-off value (EUCAST); ^3^ 2:1 ratio; ^4^ trimetoprim–sulphametoxazole in ratio 1:19.

**Table 4 vetsci-12-00691-t004:** Frequency table of minimum inhibitory concentration (MIC) values for antimicrobial agents with established clinical breakpoints in *Escherichia coli* isolates (*n* = 45) from chickens in the Dél-Alföld region. The upper row for each compound displays the number of isolates, while the lower row shows the corresponding percentage. Vertical red lines indicate the clinical breakpoint, and green lines represent the epidemiological cutoff value (ECOFF).

Antibiotics	^1^ Breakpoint	0.001	0.002	0.004	0.008	0.016	0.031	0.063	0.125	0.25	0.5	1	2	4	8	16	32	64	128	256	512	1024	MIC_50_	MIC_90_	^2^ ECOFF
	µg/mL																								
Amoxicillin	32										1	1	1	14	7	1	0	1	1	5	9	4	8	512	8
											2.2%	2.2%	2.2%	31.1%	15.6%	2.2%	0.0%	2.2%	2.2%	11.1%	20.0%	8.9%			
^3^ Amoxicillin-clavulanic acid	32												4	16	9	13	3	0	0	0	0	0	8	16	8
													8.9%	35.6%	20.0%	28.9%	6.7%	0.0%	0.0%	0.0%	0.0%	0.0%			
Ceftriaxone	4			1	1	5	6	12	6	2	1	1	0	0	1	2	1	2	1	2	1		0.063	64	0.125
				2.2%	2.2%	11.1%	13.3%	26.7%	13.3%	4.4%	2.2%	2.2%	0.0%	0.0%	2.2%	4.4%	2.2%	4.4%	2.2%	4.4%	2.2%				
Doxycycline	16												8	10	3	5	11	8					16	64	8
													17.8%	22.2%	6.7%	11.1%	24.4%	17.8%							
Enrofloxacin	2					3	3	5	0	1	5	8	4	2	5	7	1	1					1	16	0.125
						0.0%	6.7%	11.1%	0.0%	2.2%	11.1%	17.8%	8.9%	4.4%	11.1%	15.6%	2.2%	2.2%							
Florfenicol	16														8	23	12	2					16	32	16
															17.8%	51.1%	26.7%	4.4%							
Imipenem	4				1	0	11	0	3	2	9	8	6	3	2								0.5	4	0.5
					2.2%	0.0%	24.4%	0.0%	6.7%	4.4%	20.0%	17.8%	13.3%	6.7%	4.4%										
Colistin	2					2	0	5	1	11	11	4	1	0	1	0	1	0	2	0	3	3	0.5	512	2
						4.4%	0.0%	11.1%	2.2%	24.4%	24.4%	8.9%	2.2%	0.0%	2.2%	0.0%	2.2%	0.0%	4.4%	0.0%	6.7%	6.7%			
Neomycin	32														5	14	11	12	0	2	1		32	64	8
															11.1%	31.1%	24.4%	26.7%	0.0%	4.4%	2.2%				
^4^ Potentiated sulphonamide	4												1	7	4	7	1	0	8	0	7	10	128	1024	0.5
													2.2%	15.6%	8.9%	15.6%	2.2%	0.0%	17.8%	0.0%	15.6%	22.2%			
Spectinomycin	128																4	31	10				64	128	64
																	8.9%	68.9%	22.2%						

^1^ Clinical Laboratory Standard Institute (CLSI); ^2^ epidemiological cut-off value (EUCAST); ^3^ 2:1 ratio; ^4^ trimetoprim–sulphametoxazole in ratio 1:19.

## Data Availability

The data presented in this study are available from the corresponding author upon reasonable request.

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
