# Peer review of "Antimicrobial Resistance in Commensal Bacteria from Large-Scale Chicken Flocks in the Dél-Alföld Region of Hungary"

_vetsci, 2025, doi:10.3390/vetsci12080691_

Round 1
Reviewer 1 Report
Comments and Suggestions for Authors
The manuscript "Antimicrobial Resistance in Commensal Bacteria from Large-Scale Chicken Flocks in the Dél‐Alföld Region of Hungary" written by Kerek et al presented AMR profiles of Staphylococcus, Enterococcus and E. coli isolated from chicken tracheal and cloacal swabs. The manuscript extensive improvements in methodology, results, and discussion sections. The comments can be found in attached PDF document and provide revised manuscript with point-by-point response to every comment. The revised manuscript should be provided without track changes file but with highlighted text for easy tracking of revisions.

Author Response
Dear Reviewer 1,
Thank you very much for your thorough review, for pointing out the mistakes in the manuscript, and for helping to improve the quality of the manuscript with your excellent suggestions. We have corrected all the mistakes and incorporated your suggestions. The exact changes are detailed in the attached document.
Yours sincerely,
The Authors

Reviewer 2 Report
Comments and Suggestions for Authors
Great study! Please follow the below.
Line 109-110 – Could you provide with species name here?
Line 132 -The sentences is repeated earlier at line 117-119. So, remove this one.
Line 137 – Cite Coliform agar properly
Line 138-139 – I think you isolated selected colonies. Could you please add the characteristic colony morphologies of each of these bacterium on the type of chrome agar?
Figure 5 – Don’t italicize strains (line 259)
Line 441 – How did you get these clinical isolates? Is that just from literature?
Line 508-510. You need to add more information from the references. Is it a different geographical area, is it about the same type of farms, same techniques used for analysis etc. Make sure to extend this to other references used in the other paragraphs on other pathogens as well.
Line 525, 531, 536, 544 – Are you talking about a specific bacterial isolates or overall? I am noticing the same information under E. coli resistance as well. I think this is about formatting. You may add some titles for each pathogen and then add the information. Please be careful about separating information in paragraphs. And for all bacteria, if you compare your results with another study, provide more information on those studies.
Line 628 – When you add the risks of AMR and predict on consequences that may happen due to AMR, you better refer some other studies that talks about AMR is retail chicken/ meat and environmental samples from farms.
Author Response
Dear Reviewer 2,
Thank you very much for your thorough review, for pointing out the mistakes in the manuscript, and for helping to improve the quality of the manuscript with your excellent suggestions. We have corrected all the mistakes and incorporated your suggestions. The exact changes are detailed in the attached document.
Yours sincerely,
The Authors

Reviewer 3 Report
Comments and Suggestions for Authors
Merit
Antibiotic resistance continues to pose challenges in medical practice therefore studies that focus on understanding resistance patterns and evolution of resistance genes are important.
Comments
- The authors should explain how commensal bacteria may serve as reservoirs and vectors of resistance genes and clinical relevance to support the importance of the study.
- There is no need for Figure 1 as it adds no value to the manuscript. I suggest that it should be deleted. It is sufficient to state in the materials and method section that “Staphylococcus isolates originating from the Del-Alfold region were analyzed” rather than presenting entire map.
- The number of Figures is superfluous. Most of the key points can be stated in the body of the manuscript without extensive images of which some are confusing.
- The manuscript can be condensed. The length is most likely due to excessive Figures.
Author Response
Dear Reviewer 3,
Thank you very much for your thorough review, for pointing out the mistakes in the manuscript, and for helping to improve the quality of the manuscript with your excellent suggestions. We have corrected all the mistakes and incorporated your suggestions. The exact changes are detailed in the attached document.
Yours sincerely,
The Authors

Round 2
Reviewer 1 Report
Comments and Suggestions for Authors
the authors just did bacterial isolation and identification up to genus level based on colony characteristics which is not enough to rely. The authors need to perform further experiments to identify bacterial strains at special level based on biochemical, molecular, and/or genomic level using advanced technologies such as MALDI-TOF, VITEK system, PCR, 16S rDNA, and/or WGS. Moreover, the author just provided Antibiotic susceptibility profiles which I think data is not very basic and need to detect antibiotic resistance genes using PCR and/or WGS. I would like to reject this manuscript in its current form.
Author Response

(The authors gave the same response as above.)

Reviewer 3 Report
Comments and Suggestions for Authors
Comments
The authors have mostly addressed my comments from the previous submission but concerns regarding too many figures still exist.
- The figures are too many. Most of the findings can be stated in the body of the manuscript without images/Figures. The manuscript as presented makes it tedious to read and follow with ease.
- The authors did not include point by point responses to reviewer comments which could have helped in assessing the quality of the manuscript for scientific soundness.
Author Response
Dear Reviewer 3,
We are very grateful for your review. We have detailed the changes made to the manuscript in accordance with your request in the attached document.
Best regards,
The Authors
